



# Recovery and validation of Odin/SMR long term measurements of mesospheric carbon monoxide

Francesco Grieco[1], Kristell Pérot[1], Donal Murtagh[1], Patrick Eriksson[1], Peter Forkman[1], Bengt Rydberg[2], Bernd Funke[3], Kaley A. Walker[4], and Hugh C. Pumphrey[5]

[1]Department of Space, Earth and Environment, Chalmers University of Technology, Gothenburg, 412 96, Sweden
[2]Molflow, Gråbo, 443 40, Sweden
[3]Instituto de Astrofísica de Andalucía, CSIC, Granada, Spain
[4]Department of Physics, University of Toronto, Toronto, M5S 1A7, ON, Canada
[5]School of GeoSciences, University of Edinburgh, Edinburgh, EH9 3FF, UK

**Correspondence:** Francesco Grieco (francesco.grieco@chalmers.se)

**Abstract.**

The Sub-Millimetre Radiometer (SMR) on board the Odin satellite performs limb sounding measurements of the middle atmosphere to detect molecular emission from different species. Carbon monoxide (CO) is an important tracer of atmospheric dynamics at these altitudes, due to its long photochemical lifetime and high vertical concentration gradient. In this study, we have successfully recovered over 18 years of SMR observations, providing the only dataset to date being so extended in time and stretching out to the polar regions, with regards to satellite-measured mesospheric CO. This new dataset is part of the Odin/SMR version 3.0 level 2 data. The much of the level 1 dataset - except the October 2003 to October 2004 period - was affected by a malfunctioning of the Phase-Lock Loop (PLL) in the frontend used for CO observations. Because of this technical issue, the CO line could be shifted away from its normal frequency location causing the retrieval to fail or leading to an incorrect estimation of the CO concentration. An algorithm was developed to locate the CO line and shift it to its correct location. Nevertheless, another artifact causing an underestimation of the concentration, i.e. a line broadening, stemmed from the PLL malfunctioning. This was accounted for by using a broader response function. The application of these corrections resulted in the recovery of a large amount of data that was previously being flagged as problematic and therefore not processed. A validation study has been carried out, showing how SMR CO volume mixing ratios are in general in good accordance with the other instruments considered in the study. Overall, the agreement is very good between 60 and 80 km altitude, with relative differences close to zero. A positive bias at low altitudes (50 - 60 km) up to +20% and a negative bias up to -20% at high altitudes (80 - 100 km) were found with respect to the compared instruments.

## 1 Introduction

Of the carbon monoxide (CO) produced at surface level by anthropogenic sources (e.g. industrial activities, biomass burning, transport and heating) as well as by oceans and biogenic sources, very little is transported upwards to the stratosphere, as it is chemically destroyed by reacting with the hydroxyl radical (OH). This causes a sharp gradient in CO concentration from the



troposphere to the lower stratosphere (Zander et al., 1981). The CO which can be observed in the middle atmosphere is mainly produced via two processes: methane ($CH_4$) oxidation and $CO_2$ photolysis, the latter being the dominant one (Minschwaner et al., 2010). In particular, high altitudes are characterised by strong fluxes of radiation in the Schumann-Runge continuum and Lyman-$\alpha$ wavelengths which are strongly absorbed by $CO_2$, therefore $CO_2$ photolysis becomes more significant with height in

the upper mesosphere and lower thermosphere, providing a major source of CO and resulting in a large vertical gradient in its concentration. CO from high altitudes is transported downwards in the winter hemisphere polar night region due to advection and vertical eddy mixing (Solomon et al., 1985). In the mesosphere and lower thermosphere, the photochemical lifetime of CO increases with altitude from a minimum of approximately one week up to several hundreds of years. This lifetime is greater than the zonal transport time scale and of the same order of magnitude as the meridional and vertical transport time scales

in the stratosphere up to the middle mesosphere, while it is significantly greater than all transport times scales in the upper mesosphere (Brasseur and Solomon, 2005). Because of its strong horizontal and vertical concentration gradients and to its long lifetime, CO is commonly used as a tracer of middle atmospheric dynamics (e.g., Lee et al., 2011; de Zafra and Muscari, 2004). Measurements of CO in the mesosphere have been carried out since the end of the 70's using ground-based instruments (Clancy et al., 1982) and later on with satellites such as UARS/ISAMS (Allen et al., 1999). More recent satellite measurements are the

ones performed in the microwave band with Aura/MLS (e.g., Froidevaux et al., 2006), and in the infrared with Envisat/MIPAS (e.g., Funke et al., 2009) and Scisat-1/ACE-FTS (e.g., Clerbaux et al., 2008). The first retrieval results from CO measurements obtained with the Sub-Millimetre Radiometer (SMR) on board Odin were presented by Dupuy et al. (2004). They observed seasonal variations of CO concentration related to global circulation and chemical processes as predicted from the Whole Atmosphere Community Climate Model (WACCM), to which they compared their results obtaining an overall agreement within

two orders of magnitude. Also, good agreement has been obtained comparing with observations from UARS/ISAMS.

There are not many studies about Odin/SMR measurements of CO, although such measurements have been performed from August 2001 until today. This is due to a malfunctioning of the Phase-Lock Loop (PLL) in the frontend used for CO observations, which caused the majority of CO data to present artifacts which made them unusable for retrievals. The issue is illustrated in Section 2 together with a description of Odin/SMR. In Section 3 we explain how we implemented a correction

for the artifacts caused by the PLL malfunctioning and other issues which arose during the retrieval process. As a result, retrieval products from over 18 years of Odin/SMR CO observations, between 50 km and 100 km altitudes, are now available and presented in Section 4. This is the first data being part of the Odin/SMR v3.0 L2 dataset. Finally, for validation purposes, in Section 5 we compare Odin/SMR profiles with those from Envisat/MIPAS, Scisat-1/ACE-FTS and Aura/MLS, being the satellite instruments operating at the same time as SMR and observing similar altitudes. We also present comparisons with

ground-based measurements from Onsala Space Observatory.



## 2 Odin/SMR CO measurements

### 2.1 The sub-mm radiometer

The Odin satellite, a Swedish-led project in collaboration with Canada, France and Finland, was launched on 20 February 2001 into a 600 km sun-synchronous orbit with inclination 97.77° and 18:00 hrs ascending node. Its observation time was shared between astronomical and atmospheric observations until 2007, when the astronomical part of Odin's mission was concluded. After that, the instruments on board of Odin have been used exclusively for limb sounding of the atmosphere. These instruments are OSIRIS (Optical Spectrograph and InfraRed Imaging System) and SMR. In this article, we are using the latter. The four sub-mm receivers in SMR can be tuned to cover frequencies between 486 - 504 GHz and 541 - 581 GHz, thus to observe emission due to rotational transitions for species such as $O_3$, $H_2O$, CO, NO, ClO, $N_2O$, $HNO_3$ in the stratosphere and mesosphere (e.g., Frisk et al., 2003). There is also a mm receiver to observe the 118 GHz $O_2$ transition. The cold sky and a hot load are repeatedly observed for calibration purposes. Through a Dicke switch, the signal from a calibrator and the main beam signal are deviated towards different directions, and further split according to polarisation, before being collected by different receivers (see Figure 1). Every signal is thereafter converted to longer wavelengths by combining it with a local oscillator (LO) signal through a mixer. The frequency of the LO is fine tuned with the use of a PLL. The resulting signal can then be amplified and routed to the autocorrelator spectrometers (AC1 and AC2, as indicated in Figure 1).

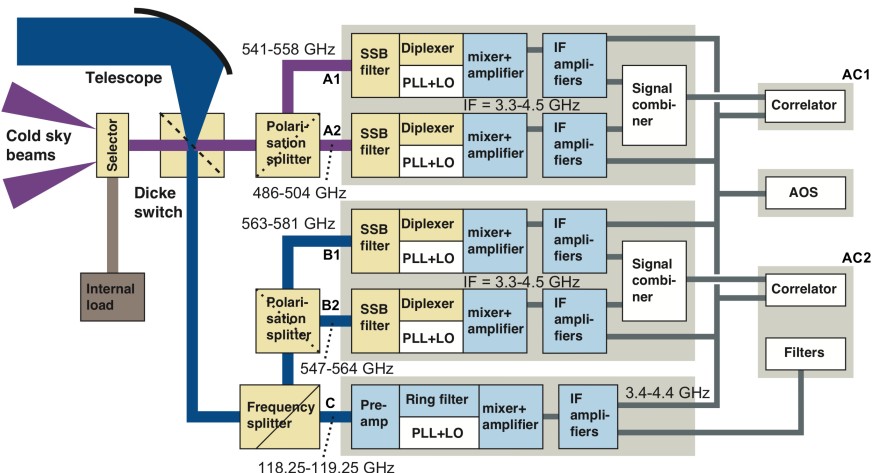

**Figure 1.** Block diagram of the Odin radiometer. From Frisk et al. (2003).

Measurements are performed during both upward and downward vertical scanning with a vertical sampling of ~6 km in the mesosphere. Odin/SMR L1 data are organised in scans, each of them consisting of a group of spectra collected during a single upward (or downward) scanning. Each spectrum corresponds to a single mean tangent altitude in the range 7 - 72 km (stratospheric scans), 7 - 110 km (strato-mesospheric scans) or 60 - 110 km (mesospheric scans) (Dupuy et al., 2004).



**Table 1.** FMs for observing CO. From Rydberg et al. (2017).

| Frontend | Spectrometer | LO Freq. [GHz] | Freq. Range [GHz] | Species | FM |
|---|---|---|---|---|---|
| 572 B1 | AC2 | 572.762 | 576.062 - 576.862 | CO, $O_3$ | 14 |
| | | 572.964 | 576.254 - 576.654 | CO, $O_3$ | 22 |
| | | | 577.069 - 577.469 | $HO^2$, $^{18}O_3$ | |
| | AC1 | 572.762 | 576.062 - 576.862 | CO, O3 | 24 |

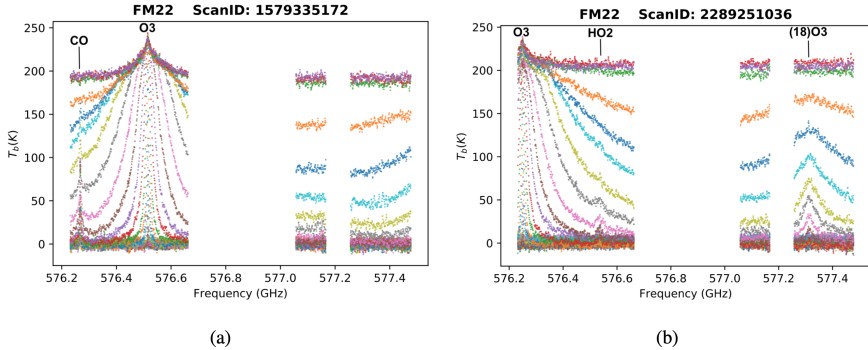

(a)                                                     (b)

**Figure 2.** Spectra for altitudes ranging from 7 km to 110 km for two scans taken as examples. The bigger gap in the spectra around 576.8 GHz is due to the particular way the bands are arranged to cover the desired frequencies for FM22, while the smaller gap at 577.2 GHz is due to instrumental failure relative to one sub-band. In (a) both CO and $O_3$ lines are present and in the right place. In (b) the frequency shift is so big that only the $O_3$ can be observed (shifted) while the CO line is shifted outside of the band and therefore not observed. Colors corresponds to different tangent altitudes within the reported scan.

## 2.2 SMR CO operational modes

Odin/SMR measures CO, with a 3 - 4 km vertical resolution, from the thermal emission line corresponding to the $J = 5 \rightarrow 4$ rotational transition at 576.268 GHz. In Figure 2a we show an illustrative scan of spectra obtained in normal conditions where both a CO line and a $O_3$ line are measured. The SMR receivers can be adjusted to different configurations called frequency

5    modes (FM), each of which corresponds a certain observed frequency band and a scheduled observation time. There are three FMs that cover bands in which the above mentioned CO transition is observed. Their characteristics are summarised in Table 1. All these measurements are carried out by using the B1 frontend (the set of components thus denoted in Figure 1), whose PLL element was working correctly for a year only, between 8 October 2003 and 8 October 2004. The measurements made during the rest of the Odin operational time, from 2001 until today, have all been affected by a malfunctioning of this PLL, leading to a

10    shift of the LO frequency from its nominal value (Rydberg et al., 2017). Consequently each scan presents a different frequency shift. In extreme cases like the one shown in Figure 2b, the frequency shift causes the CO line to fall outside the observed bandwidth and thus the data are unrecoverable.



## 3 Recovery and retrieval

We explain here how we developed a frequency correction which was applied to the spectra before the retrieval process, leading to the recovery of a great part of the dataset. This is followed by descriptions of the set-up for the v3.0 CO retrievals and of the method used to estimate line broadening generated by the PLL failure.

### 3.1 Basic frequency correction

In order to correct the frequency shift, as a first step, the centre frequency of the CO line in the scan's average spectrum has been compared with the theoretical centre frequency. The resulting difference is then applied to the LO frequency. To do that, the correction algorithm first needs to distinguish the CO line from the $O_3$ line. In Figure 3 it is shown how peak brightness temperatures of CO and $O_3$ lines vary with altitude. Here it can be noticed how in the 40 - 60 km altitude range the CO and $O_3$

10 slopes are most different, allowing us to discern between the two species. Averaging of observations during the PLL working period show that a -0.0004 ± 0.001 K/m slope corresponds to the CO line, while the value -0.009 ± 0.003 K/m identifies the $O_3$ line, where the uncertainties on slopes correspond to $3\sigma$. Thus, the correction algorithm associates peak $T_b$ gradients with a value higher than -0.0045 K/m to the CO line. If no line or only the $O_3$ line is found, the scan is not considered for further processing.

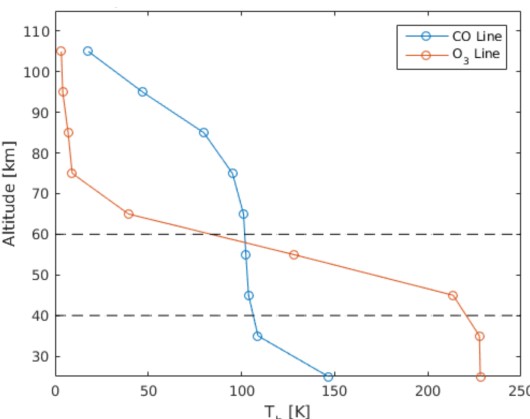

**Figure 3.** Peak brightness temperatures of CO and $O_3$ at different altitudes. Average considering all observations performed during the PLL working period, i.e. 14224 scans. At altitudes between 40 km and 60 km, the CO line curve presents a steeper slope.

15 Despite the application of this first correction to the data, they still present artifacts that need to be corrected. In fact, as can be seen in Figure 4a, all spectra in a single scan, each corresponding to a different altitude, present a different frequency shift. This suggests that the PLL malfunctioning affects the observation at a time scale smaller than the scanning time scale (~2 minutes). Thus applying the same correction to each spectrum in a scan is not sufficient. To solve this problem, we modified the pre-correction algorithm by considering each single altitude spectrum and estimating the observed center frequency of





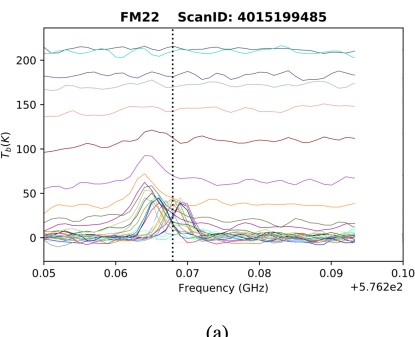
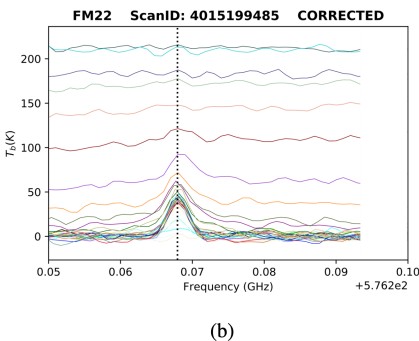

(a)                                    (b)

**Figure 4.** Spectra measured on 2 February 2009 presenting altitude dependent frequency shifts before (a) and after (b) altitude dependent correction. Colors corresponds to different tangent altitudes within the reported scan.

each CO line by fitting them with a Gaussian. We then compared these values to the theoretical center frequency and therefore applied a different shift to each spectrum. An example of result obtained from this correction is shown in Figure 4b. Including this altitude dependent correction resulted in a significantly increased amount of data that could be recovered, i.e. from which CO concentration profiles could be derived (as shown in Figure 7 in Section 4) .

## 3.2 Retrieval set-up

A new version of the ARTS operational processing system has been developed for SMR retrievals. The data produced by this system will be denoted as version 3.x. The new retrieval system will be described in detail in a forthcoming publication, but a summary is provided below. The CO data discussed in this work have been assigned version number 3.0.

The new processing system is based on a MySQL database and is capable of distributing the calculations over multiple clusters. The actual retrievals are based on the Atmospheric Radiative Transfer Simulator (ARTS, Buehler et al. (2018)). For the moment ARTS version 2.3.564 is applied. Besides atmospheric radiative transfer, the forward simulations consider the sensor's antenna response, double-sideband characteristics and spectrometer frequency response, using the approach of Eriksson et al. (2006). ARTS provides the needed weighting functions (mainly using analytical expressions), while the inversion of optimal

estimation type is made in Matlab by code taken from Eriksson et al. (2005).

The L2 data provided contains a characterisation of the retrieval, following Rodgers (2000). As described in Baron et al. (2002), when retrieving multiple quantities in parallel, the "smoothing error" includes terms that describe the interference between the quantities. The retrieval error reported in SMR v3.0 data incorporates this cross-quantity interference, but excludes the classical smoothing error internal to the quantity. For example, the error reported for CO considers that the retrieval of in-

strumental parameters is non-ideal, but excludes the direct smoothing of the CO profile due to limitations in vertical resolution. In this way, the reported error matches the standard one that would be obtained if the instrumental parameters would instead be treated as forward model uncertainties (though strictly true only for a totally linear inversion problem).



Specifics for the retrievals presented in this paper include that only spectra recorded at tangent altitudes between 40 and 100 km are considered, to form a "mesospheric retrieval mode". The spectroscopic data for the frequency range of concern is solely taken from HITRAN 2012 (Rothman et al., 2013) (but the system allows to incorporate data from other sources). The set of variables retrieved is specified for each mode separately. In this case, the CO profile and off-sets for pointing, frequency and

5 brightness temperature zero level (baseline) are retrieved.

The a priori dataset previously used for Odin/SMR CO retrievals was found outdated and inaccurate. As a consequence, for the new inversions, a new CO climatology was formed, based on MIPAS zonal means averaged over the years separately for each month. For this purpose, we are using the product V5R_CO_521, which at the moment is the most recent data version from MIPAS middle atmosphere observation mode (Garcia et al., 2014) (see Section 5.1).

In Figure 5 is shown the retrieval for an exemplary scan. Despite spectra above 40 km tangent altitude are considered in the inversion, only measurements above 50 km are reliable since at lower altitudes it is not possible to discern the CO line from the noise. Moreover, data with a measurement response lower than 0.75 are discarded. This is the case of the retrieved profile above 100 km which is dominated by the a priori and is here shown out of completeness, together with the averaging kernel extending to higher altitudes, to display how retrieved concentrations between 90 and 100 km are also influenced by higher

altitudes.

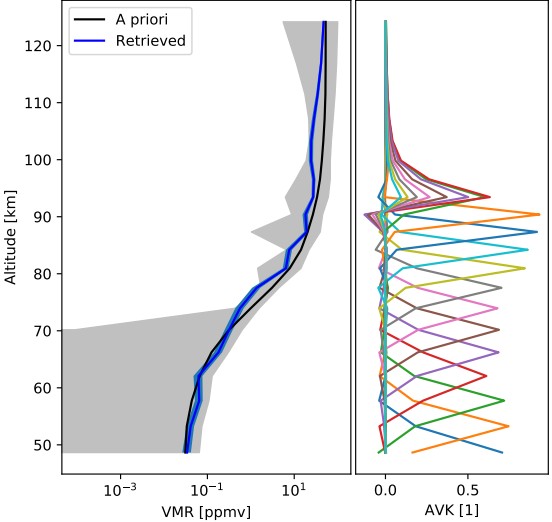

**Figure 5.** Example retrieval referring to ScanID 1579335172 from FM 22. Left: Retrieved concentration profile and error (in blue) and a priori including uncertainties (in grey). Right: Averaging kernel plotted in a different colour for each altitude (not indicated).





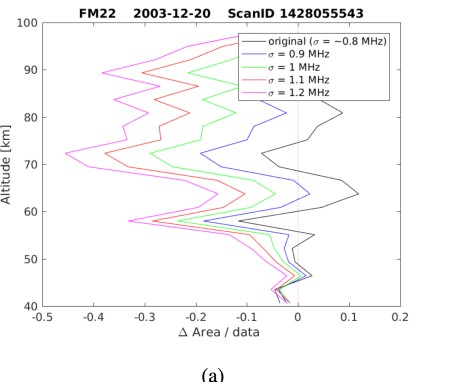 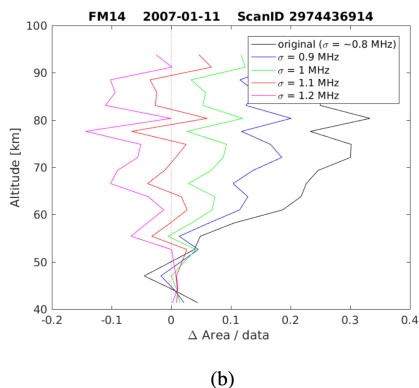

(a) (b)

**Figure 6.** Relative difference between the Gaussian area beneath the observed CO line and the Gaussian area beneath the fit, obtained with different response functions for a scan in the PLL working period (a) and non-working period (b). The best response functions for the two cases are, respectively, the original one (black line) and the one with $\sigma = 1.1$ MHz (red line).

### 3.3 Broadening correction

The fits of CO line performed during the inversion process - which are Gaussian shaped due to Doppler broadening - have significantly different amplitude and width compared to the observed line, causing an underestimation of the concentration. This suggests that the PLL malfunctioning even has an effect within the integration time, causing the observed line to be
broadened with respect to what it would be in normal instrumental conditions. To take into account this broadening issue during the inversion, the original response function, a Gaussian with $\sigma = 0.8$ MHz, has been replaced with a Gaussian with a higher $\sigma$ value. The response function $w_{ch}^i(\nu)$ of a spectrometer's channel $i$ represents how the power of a signal $I(\nu)$ is weighted:

$$y_i = \int_0^\infty I(\nu)w_{ch}^i(\nu)d\nu \quad \text{with} \quad \int_0^\infty w_{ch}^i(\nu)d\nu = 1 \tag{1}$$

where $y_i$ is the final, calibrated antenna temperature of the channel $i$ (Eriksson et al., 2006).

In Figure 6a is represented the relative difference between the Gaussian area beneath the observed CO line and the Gaussian area beneath the fit, for a scan in the period when the PLL was working. Here there is no need for correction, and the lower values of the relative area difference are obtained with the original response function. As an example, in Figure 6b is plotted the relative area difference for one scan in the PLL malfunctioning period. During this period, the most suitable response function
is a Gaussian with $\sigma = 1.1$ MHz. This is indeed the one resulting in the lowest overall relative area differences in the whole period after 8 October 2004.

Note that the original response function is still the one giving the best results during the period before 8 October 2003, probably because the PLL malfunctioning in the very beginning of the satellite operation time, before 8 October 2003, is of



different nature than the one after 8 October 2004. Consequently, the new response function has been applied only to inversions after 8 October 2004.

## 4   The new dataset

The recovered dataset is part of the Odin/SMR v3.0 L2 data. As mentioned in Section 3, the new frequency correction algorithm also helped to recover many data that were previously being erroneously flagged, so that there is a very significant increase in the amount of CO level 2 data that are now available. This can be seen in Figure 7, showing the temporal distribution of the L1 and L2 data sets before and after the application of the correction algorithm, for the three CO observation modes together. The results are very satisfying, with 63% of the data being successfully processed (compared to 8% before the correction) until

September 2017. The remaining data are discarded due to other quality flags. After this date, the frequency shift is causing the CO line of almost all the spectra to be outside of the observed band, resulting in the loss of most of the CO data during this time period, hence the almost total absence of L2 data from there on. The very high number of measurements performed with CO FMs during July 2002, July 2003 and August 2004 corresponds to special scheduling of the observation time, set to monitor dynamics in the northern summer mesosphere associated with the study of noctilucent clouds (Karlsson et al., 2004).

The time series shown in Figure 8 gives an overview of the new CO data set, extended to cover the whole globe until the polar regions. They consist of monthly zonal means of CO volume mixing ratios over five different latitude bands. The white bands correspond to months during which the number of scans in the given latitude band is lower than 10. No concentrations are shown for some months in the early years of the mission, despite the presence of sufficient L2 data, because of plotting

interpolation with adjacent months with no data (see Figure 7). No significant difference between the various FMs is observed (not shown). CO volume mixing ratios show noticeable variations with altitude, latitude and season, as well as longer-term variations. We observe a sharp increase with height, due to the photodissociation of $CO_2$ at high altitudes, as explained in Section 1. In these plots, it is also possible to notice the temporal variation of CO in the mesosphere throughout the years. In the tropics, maxima appear at equinoxes and minima at solstices. This is explained by the Semiannual Oscillation (SAO)

signal, which dominates at low latitudes (Lee et al., 2018). At high latitudes, the seasonal variations are mainly characterised by the downward transport of CO rich air in winter, from the upper mesosphere to the stratopause, induced by the meridional circulation. The CO volume mixing ratio values measured in summer are lower, due to upward circulation. The SAO signal is noticeable in this latitude band too, explaining the secondary minima that are visible in the middle of the winter (Lee et al., 2018). The seasonal variations at mid latitudes are similar, though the effect of the meridional circulation is significantly less

pronounced than at high latitudes. Moreover, at high latitudes in the northern hemisphere, a particularly strong secondary peak in CO concentration appears during several of the winters. This is observed in particular in 2006, 2009, 2013 and 2019. As seen in Figure 8, unusually high volume mixing ratio values were measured in the mid and lower mesosphere, in those years, in late winter. Such a pattern is generated by Sudden Stratospheric Warming (SSW) events followed by an elevated stratopause




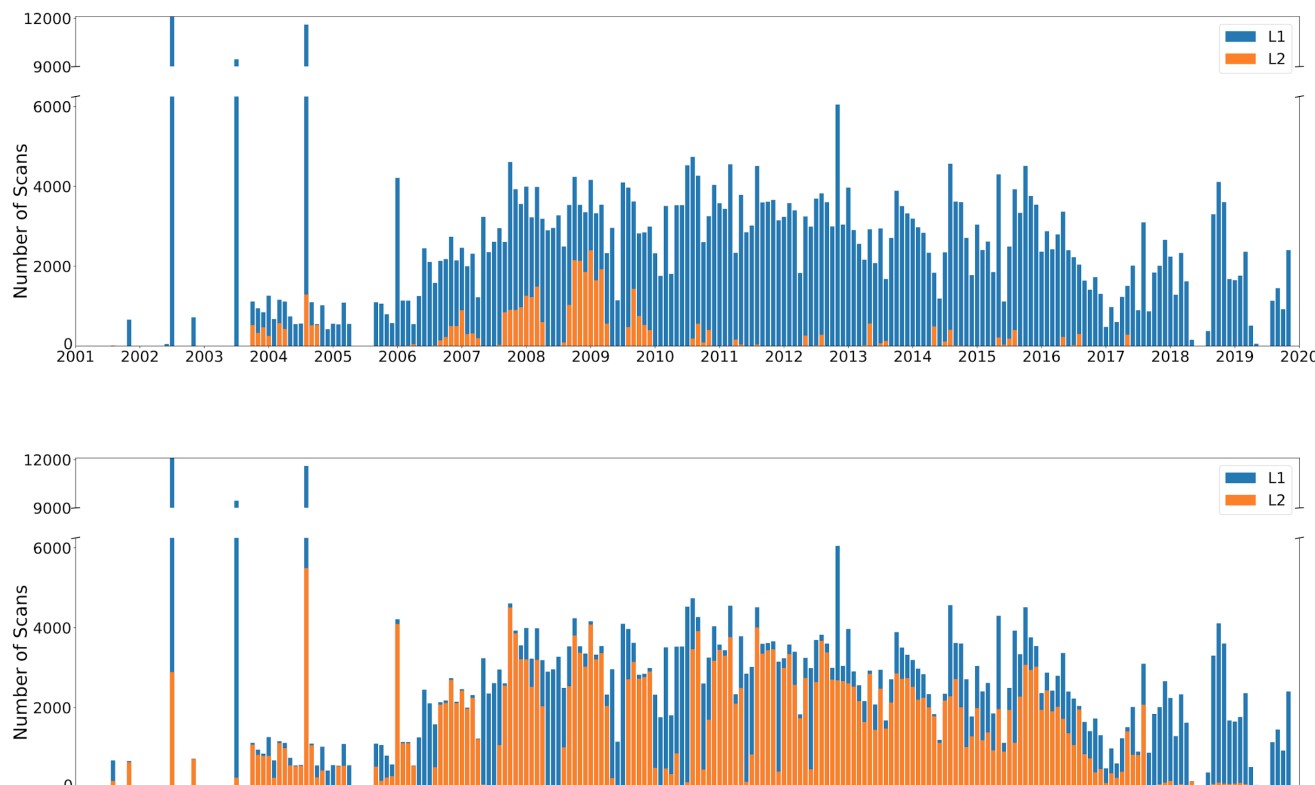

**Figure 7.** Number of L1 and L2 scans by month for FM14, FM22 and FM24 altogether, before (top) and after (bottom) the application of the correction algorithm. The ticks on the x-axis correspond to January 1st for each year. This figure gives an overview of how much data could be recovered.

event. SSWs occur mainly in the northern hemisphere, during wintertime, and consist of warmings of the stratosphere at high latitudes by several tens of Kelvins, occurring within a few days. This is caused by planetary waves disturbing the polar vortex (e.g., Charlton and Polvani, 2007), thus reducing the descent of CO rich - air from higher altitudes. After such an event, the vortex recovers, and the stratopause sometimes reforms at higher altitudes than normal (Vignon and Mitchell, 2015), as it has

5 already been observed by Odin/SMR (Pérot et al., 2014). In such a case, the downward transported air will come from higher altitudes (Orsolini et al., 2017), where CO is more abundant, resulting in the above mentioned higher concentration peak in late winter. Finally, it can be seen that, in all latitude bands, concentrations are in general higher during the period 2012-2016, in accordance with corresponding more intense solar activity. The described temporal variations are consistent with observations from other satellite instruments, such as MLS (Lee et al., 2018) and MIPAS (Funke et al., 2009; Garcia et al., 2014).

**Figure 8.** Time series of CO volume mixing ratios measured by SMR for different latitude bands. The white bands indicate periods during which the number of scans in the given latitude band is lower than 10. The ticks on the x-axis correspond to the beginning of each year.

## 5 Comparison with other instruments

In this section we compare the SMR v3.0 CO data set with data from other limb-sounding satellite-borne instruments, namely MIPAS, ACE-FTS and MLS, and with ground-based measurements made by a radiometer located at Onsala Space Observatory (OSO). Our goal is to assess the quality of the new data set described above. With regards to satellite measurements, the comparison is performed between measurements that occurred within a maximum temporal separation of 24 hours and a maximum spatial separation of 500 km. It is possible to use such broad coincidence criteria because of CO's long chemical lifetime in the mesosphere (Minschwaner et al., 2010). CO in the mesosphere and lower thermosphere (MLT) is affected by tidal mixing (Garcia et al., 2014) which might contribute to the biases observed in comparisons with such a broad temporal





coincidence criterium. To investigate if this is the case, we also carried out comparisons with 3h and 6h coincidence criteria. When not characterised by the absence of sufficient coincidences, these comparisons do not present biases which are significantly different from the 24h ones. No plot is shown for the 3h and 6h comparisons. All the validation plots shown hereafter refer to 24h coincidences.

CO vertical concentration profiles do not feature particular structures which would justify taking into account the satellite instruments' different vertical resolutions, also considering that such differences are not as marked as the ones between SMR and OSO (coincidence criteria and smoothing process for the SMR-OSO comparison are presented in Section 5.4). The comparison between space-borne instruments has therefore been performed simply by linearly interpolating each coincident profile over a common 1km altitude grid, ranging from 50 to 100 km. Given a couple of coincident measurements denoted with $i$, the

absolute difference of the two at altitude $z$ will be given by:

$$\delta_{abs,i}(z) = x_{SMR} - x_{comp} \tag{2}$$

and the relative difference:

$$\delta_{rel,i}(z) = \frac{x_{SMR} - x_{comp}}{(x_{SMR} + x_{comp})/2}, \tag{3}$$

where $x_{SMR}$ and $x_{comp}$ are respectively the CO mixing ratios measured from SMR and the comparison instrument at altitude $z$

for the coincidence $i$. The relative difference has as its denominator the mean of the two concentrations. This is done because, being both satellite measurements, they can both be affected by large uncertainties and none of the two is preferable as a reference (Randall et al., 2003). To minimise the weight of outliers, the average difference $\Delta(z)$ over all the $N(z)$ coincidences at altitude $z$ is calculated as a median. The dispersion of the results is represented by the standard deviation of the median, calculated as follows:

$$SEM(z) = \frac{1}{\sqrt{N(z)}} \sqrt{\frac{1}{N(z)-1} \sum_{i=1}^{N(z)} (\delta_i(z) - \Delta(z))^2}. \tag{4}$$

This is valid for both absolute and relative difference. The average and standard error of the concentration profiles are calculated using the same method. For the sake of clarity, in the following subsections we only show vertical profiles averaged over all the found coincidences, regardless of time or location. However, figures showing the relative differences between SMR and the other limb sounders as a function of altitude and latitude, for each season, has been included in the appendix and will be

regularly referred throughout the text.

## 5.1  MIPAS

The Michelson Interferometer for Passive Atmospheric Sounding (MIPAS) is a mid-infrared spectrometer which was launched on board Envisat on 1 March 2002 and was operating until April 2012, when an unexpected loss of contact with the satellite



occurred. The satellite travelled at 800 km altitude on a sun-synchronous orbit with a 98.55° inclination and a 22:00 hrs ascending node. The concentration profiles that we use for comparison in this study are retrieved from measurements of $^{12}C^{16}O$ roto-vibrational emissions in the $v = 1 \rightarrow 0$ band around $4.7\mu m$, using the retrieving processor developed at the Institute of Meteorology and Climate Research (IMK) in Karlsruhe and the Instituto de Astrofisica de Andalucia (IAA) in Granada (e.g., 5 Funke et al., 2009; Sheese et al., 2016). The forward model used for CO retrieval takes into account non-LTE effets. We consider data from the most recent MIPAS datasets, as specified in Table 2 (Garcia et al., 2014, 2016). All recommendations about the quality filtering of the data have been followed (Kiefer and Lossow, 2017).

**Table 2.** Characteristics of the MIPAS CO datasets used for comparison. Vertical resolutions refer to the observations in the altitude range 50 - 100 km considered in this study.

| Observation Mode | Altitude Range | Vertical Resolution | Spectral Resolution Mode | Time Period | Version |
|---|---|---|---|---|---|
| Nominal (NOM) | 10 - 70 km | 5 - 15 km | Full Resolution (FR) | July 2002 → March 2004 | V5H_CO_20 |
| | | 6 - 16 km | Optimized Resolution (OR) | January 2005 → April 2012 | V5R_CO_220 |
| Middle Atmosphere (MA) | 20 - 100 km | 5 - 15 km | | | V5R_CO_521 |
| Upper Atmosphere (UA) | 42 - 150 km | 5 - 12 km | | | V5R_CO_621 |

### 5.1.1 Nominal mode

There are two observation modes called "Nominal": one in use during the period before the interferometer in MIPAS started 10 malfunctioning, when the instrument was being used in full spectral resolution (FR mission); and one in use after the instrument was recovered and brought back to function with a reduced spectral resolution (OR mission) (Oelhaf, 2008). Both FR-NOM and OR-NOM datasets are considered here altogether. With the above mentioned coincidence criteria we found 94989 coincident measurements with SMR over the July 2002 - April 2012 period during which MIPAS was operating. The profiles and differences between 50 and 70 km altitude, averaged over the whole time period and over the whole globe, are shown in Figure 15 9a. SMR difference remains positive and almost constant with altitude, staying below +15%. The main contribution to this difference comes from latitudes between -25° and +50°. In particular during northern summer, around +25°, where it reaches values above +40 % between 65 and 70 km. Everywhere else relative difference values remain within ±20% (see Figure A1).

### 5.1.2 Middle atmosphere mode

Another observation mode in use during the OR mission is the middle atmosphere mode, covering a larger altitude range. Its 20 average comparison with SMR over 17003 coincidences is shown in Figure 9b. Both data sets are in general in good agreement with each other. It can be seen that the average difference always stays between -20 % and +20%. In particular, SMR presents a positive difference of around +20% at 50 km which decreases with altitude until it becomes null at 60 km. The difference keeps being negligible up until 70 km, then it becomes negative and keeps decreasing until it reaches a minimum of -20% around 85 km after which it goes back up to +5% at 100 km. When looking at latitudes separately, the relative differences are mainly



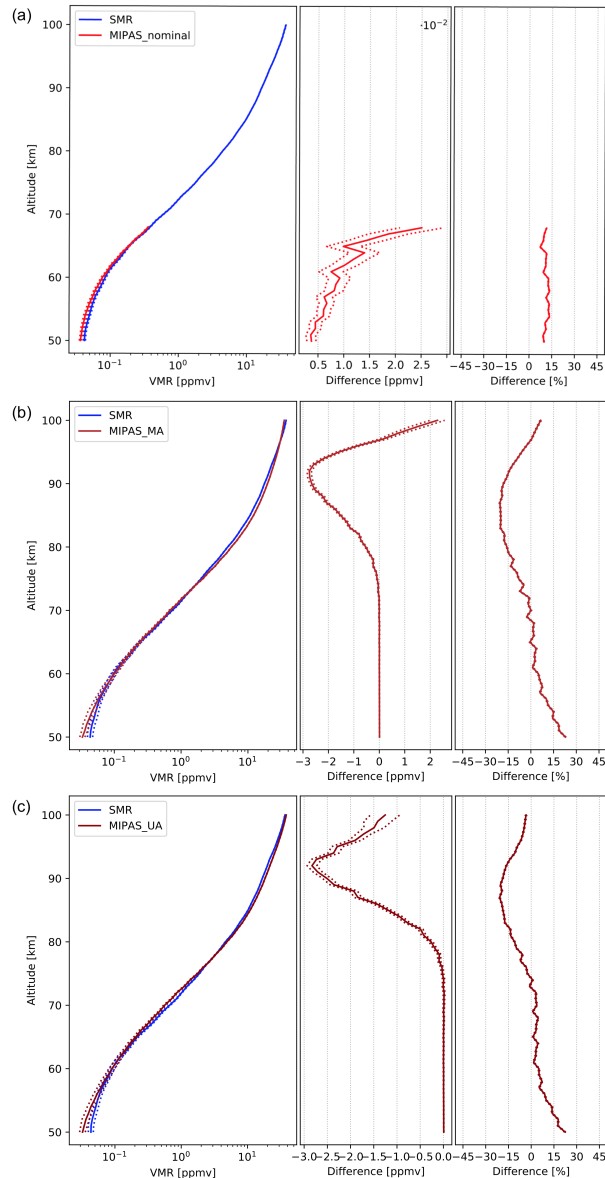

**Figure 9.** Comparison of SMR CO concentrations with the ones from MIPAS nominal (a), middle atmosphere (b) and upper atmosphere (c) modes. The data plotted are global averages over the whole time periods indicated in Table 2. Left panels: volume mixing ratios, expressed in ppmv. Center panels: absolute differences, expressed in ppmv. Right panels: relative differences, expressed in percentage. The dashed lines represent the standard deviation of the median which, in some cases, is smaller than the thickness of the profile line, causing the dashed line not to be distinguishable.



within $\pm20\%$. The most extreme differences are observed around the northern spring equinox. Here we have very negative differences at the equator between 70 and 80 km altitude reaching peaks of -60%. -50% differences can be observed during local summer in both hemispheres at high latitudes between 80 and 90 km. Also peaks of positive difference around +60% are reached at 50 km altitude at -75° and +50° (see Figure A2).

### 5.1.3 Upper atmosphere mode

Figure 9c shows the comparison with OR upper atmosphere mode, averaged over 19084 coincidences. SMR difference in comparison with this mode is very similar to the one described in Sect. 5.1.2. A small dissimilarity is given at higher altitudes where the difference approaches zero but stays negative. At all latitudes and during all seasons, relative difference values are small and generally within $\pm20\%$. Peaks of almost -50% are reached during the northern spring at the equator between 70 and 80 km and during local summer in both hemispheres at high latitudes between 80 and 90 km (see Figure A3).

### 5.2 ACE-FTS

The Fourier Transform Spectrometer (FTS) is an instrument which is part of the Canadian-led Atmospheric Chemistry Experiment (ACE) on board Scisat-1, launched on 12 August 2003 (and still operating) into a 74° inclination orbit at 650 km altitude (Bernath et al., 2005). ACE-FTS performs solar occultation measurements in the mid-infrared. In particular, CO concentrations are retrieved from the roto-vibrational absorption bands around $4.7\mu m$ (v $= 1 \rightarrow 0$) and $2.3\mu m$ (v $= 2 \rightarrow 0$). CO retrievals are performed between 5 and 105 km altitude, with a 3-4 km vertical resolution (Boone et al., 2005, 2013). In this study we use ACE-FTS v3.6 dataset for comparison. The data has been quality filtered according to the guidelines from the instrument team (Sheese et al., 2015).

A total of 12925 coincident CO measurements between ACE-FTS and SMR were found between February 2004 (first available ACE-FTS CO measurements) and March 2019. The number of coincidences found is lower compared with any of the MIPAS modes, despite the longer period considered. This is due to the fact that ACE-FTS is a solar occultation instrument, thus scanning the limb only twice per orbit, while both SMR and MIPAS are measuring almost continuously. The spatial and temporal average comparison with ACE-FTS is shown in Figure 10. The relative difference decreases almost monotonously with altitude, from a maximum of +15% at 50 km to a minimum of almost -15% at the highest altitudes. In particular, the difference decreases more quickly between 50 and 60 km from 15% to near 0%. SMR and ACE-FTS are in extremely good agreement with each other between 60 and 75 km, with a relative difference close to zero. The difference then slowly decreases with altitude up to about -15% at 90 km, and finally increases almost imperceptibly between 90 and 100 km. Looking at seasons separately, the relative differences present generally low values, almost always within $\pm20\%$. Values only increase to more than +40% between 60 and 70 km during northern autumn around +25°, and during southern winter at the equator and around -25°. The latter difference is similar to the one observed from ACE with respect to MIPAS (Sheese et al., 2016). Moreover, strong negative differences of about -40% are observed at the equator at 80 km altitude during northern spring, and also at +25° during northern autumn between 80 and 90 km (see Figure A4). Note that the majority of ACE-FTS measurements





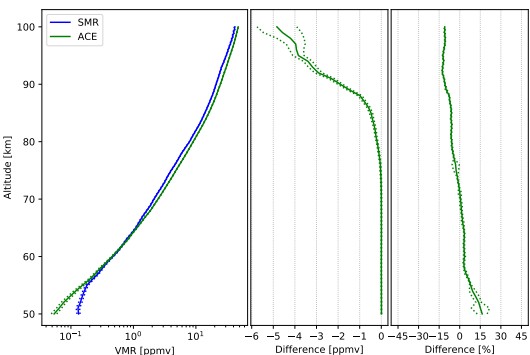

**Figure 10.** Comparison of SMR CO concentrations with the ones from ACE-FTS retrievals. The data plotted are global averages over the whole time between February 2004 and March 2019. Panels characteristics are the same as in Figure 9.

occur at latitudes higher than 60° and that the above mentioned differences at low latitudes are probably related to averaging on a smaller number of coincidences.

## 5.3 MLS

The Microwave Limb Sounder (MLS) is an instrument on board the Aura satellite, still operating since its launch on 15 July 2004. Aura/MLS was launched on a 98° sun-synchronous orbit with 13:45 hrs ascending node at an altitude of 705 km. It performs measurements between 118 GHz and 2.5 THz with a 1.5 - 3 km vertical resolution (Schoeberl et al., 2006; Waters et al., 2004). In this study we use MLS CO concentrations from the v4 dataset, retrieved from the $J = 2 \rightarrow 1$ rotational transition emission line at 230.5 GHz, which are considered to be reliable between 0.0046 hPa and 215 hPa. Also, the suggested quality filtering has been performed (Livesey et al., 2018). A previous comparison between SMR and MLS CO retrievals was presented in Barret et al. (2006), based on older versions of the datasets. A validation study of the MLS CO v2 dataset has been performed by Pumphrey et al. (2007) and the differences between v2 and v4 datasets are documented in Livesey et al. (2018).

Because MLS uses the same observation technique as SMR, and because it has been functioning since 2004, a great number of coincident measurements could be used for this validation study. Indeed, 227820 coincidences were found between the beginning of MLS mission and March 2019. The overall average comparison of these measurements with the ones from SMR is shown in Figure 11. A negative difference of SMR with respect to MLS characterizes all of the altitude range. The relative differences are oscillating between -40% and -10% and are characterised by a significant variability. This is due to MLS CO profiles being rather jagged, as reported in Errera et al. (2019) where the described MLS bias is in accordance with what we obtain in this study. Also, the MLS–ACE bias which is reported in Sheese et al. (2016) - although referred to older versions of the two datasets - is consistent with the one we measure. The difference we measure is varying significantly with latitudes and seasons: between -40° and +40° and below 70 km, there are peaks of -150% and -80% during northern spring and autumn,





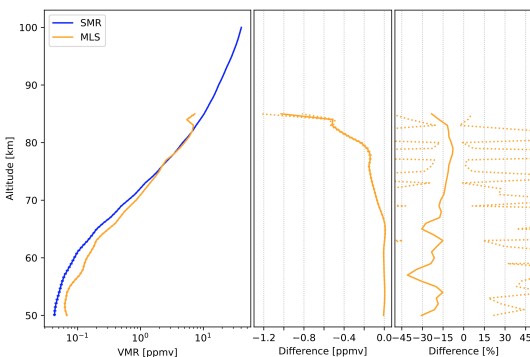

**Figure 11.** Comparison of SMR CO concentrations with the ones from MLS observations. The data plotted are global averages over the whole time between July 2004 and March 2019. Panels characteristics are the same as in Figure 9.

respectively; while marked negative differences are registered during local summertime in both hemispheres, getting more pronounced toward high latitudes, especially around 50 km and 70 km altitude with peaks of -120%, as shown in Figure A5 (this is also observed in the ACE–MLS comparison in Sheese et al. (2016)).

## 5.4 OSO

Odin/SMR CO data are compared with data measured by a ground-based remote-sensing instrument at Onsala Space Observatory, OSO (57.4° N, 11.9° E), during 2002–2007. The recommended quality filtering has been followed.

The OSO instrument is a frequency-switched microwave radiometer for observations of the CO 1 → 0 transition at 115.27 GHz. During 2002–2007, a cooled Schottky single sideband mixer was used as the first stage. In 2014 the instrument was modified to a double sideband system with a low noise amplifier, at ambient temperature, as the first stage. During 2002–2007 a spec-
trometer with 20 MHz bandwidth and a resolution of 25 kHz was used. The Optimal Estimation Method together with the forward model ARTS, the Atmospheric Radiative Transfer Simulator (Eriksson et al., 2011), has been used to retrieve vertical CO profiles from the measured spectra. The OSO instrument, the calibration and retrieval methods are described in Forkman et al. (2012, 2016).

2002–2007 daily averages of CO profiles are retrieved from the OSO spectra. Only the 24h average CO spectra with a signal
to noise ratio larger than 2 are used in the retrievals. SMR data are regarded coincident to OSO if the differences in latitude and longitude do not exceed ± 5° and ±10° respectively and if the SMR data are taken within the OSO time average periods mentioned above. For the period 2002–2007, there are 89 coincident SMR profiles, see Figure 12.

Figure 13 shows the mean vertical CO profiles for the coincident SMR and OSO data sets. The limb sounding SMR has a much higher vertical resolution than the upward-looking OSO instrument. To compensate for this difference, the SMR profiles,

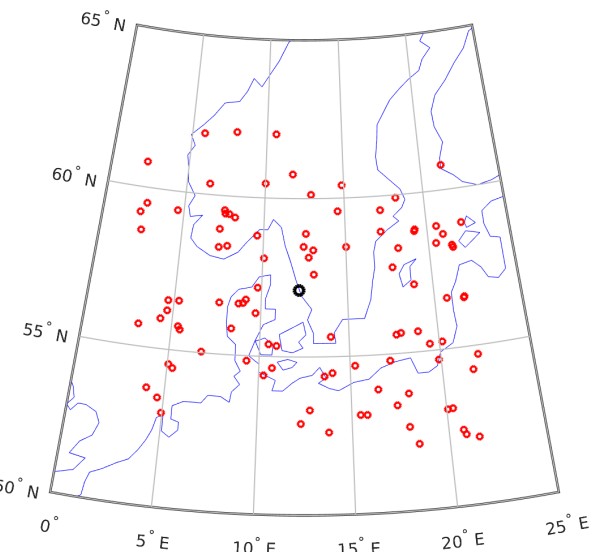

**Figure 12.** Coincidences Odin/SMR–OSO. Red circles show the 89 coincidences during 2002–2007 and the black circle shows the OSO site.

$x_{sat}$, were convoluted with the averaging kernels, $A$, of the OSO instrument (Rodgers, 2000).

$$x_s = x_a + A(x_{sat} - x_a) \tag{5}$$

where $x_a$ is the OSO apriori and $x_s$ the smoothed SMR profile. The measurement response of SMR is very close to 1 in the 50–100 km range. The altitude range in Figure 13 is more narrow since it is given by the range where the measurement

response of the OSO instrument is $> 0.75$.

OSO data from 2002–2007 have been compared to data from the satellite instruments ACE-FTS, MIPAS and MLS in a previous study (Forkman et al., 2012). As seen in Figure 13, during this period, the difference between SMR data and the coincident averaged data from OSO was found to be of 30% at 50 km, decreasing with altitude to reach -10% at 60 km and remaining constant up until 90 km altitude. The standard error of the mean, $\sigma_{\overline{x}}$, for the average difference between SMR and

10 OSO, is about 5 % above 55 km, see Figure 13.

## 6   Conclusions

Before the application of the corrections described in this study, almost the whole Odin/SMR CO dataset was unusable due to line shifts and broadening of instrumental origin, due to the Phase-Lock Loop malfunctioning. Line displacement resulted in the failure of inversions or inaccurate retrievals, while the instrumental broadening caused underestimation of the concentration

values. We estimated and corrected the different impacts of the PLL malfunctioning on SMR CO measurements. Line shifts were addressed by developing a correction algorithm which allowed the CO lines to be re-positioned to their theoretical center, also considering different frequency shifts for each tangent altitude within a scan. This resulted in the recovery of a great part of





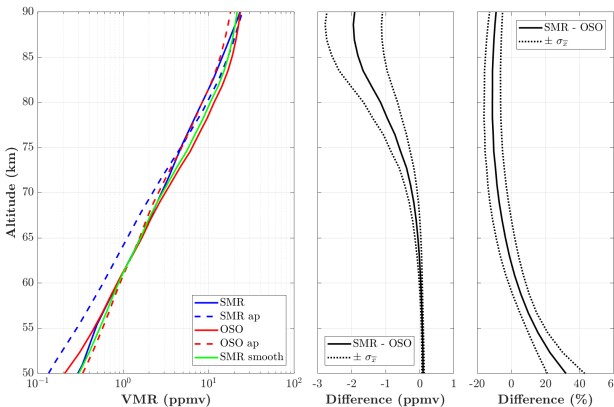

**Figure 13.** SMR–OSO comparison 2002–2007. Left: blue solid/dashed are the average SMR retrieved/apriori profiles, red solid/dashed are the average OSO retrieved/apriori profiles and green is the smoothed SMR profile. Middle and right: the average difference between the smoothed SMR and OSO profiles, solid, together with the standard error of the mean, $\sigma_{\overline{x}}$, dotted, are shown in ppmv and %, respectively.

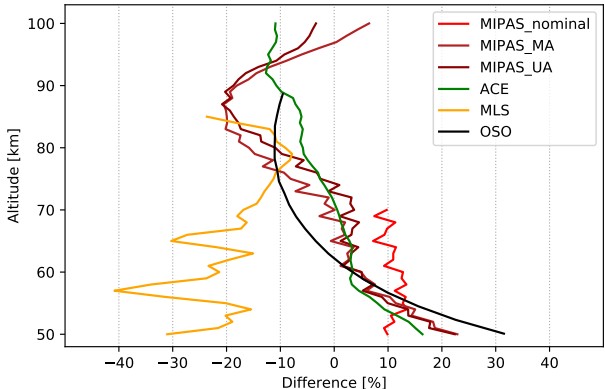

**Figure 14.** Summary of relative differences between SMR CO concentrations and the ones measured by all other instruments considered in this study. For the sake of clarity, errors are not shown.

the dataset. Line broadening was taken into account using a broader response function for the retrievals. That led to a new, good quality, Odin/SMR v3.0 mesospheric CO dataset, covering more than 18 years of observations. Time series of the retrieved volume mixing ratios reveal variations consistent with known mesospheric dynamical patterns, such as the annual cycle of the meridional circulation, SAO, SSWs, as well as a clear 11-year solar cycle signal. The validation study shows, on average, a good agreement with both the ground-based radiometer OSO and the three satellite-borne instruments (MIPAS, ACE-FTS, MLS) considered for comparison (see Figure 14). In particular, between 60 and 80 km SMR agrees very well with almost all instruments, presenting relative differences close to zero. Comparisons with MIPAS, ACE-FTS and OSO show a positive bias of SMR of up to +20% at low altitudes (50 - 60 km) and a negative bias of up to -20% at high altitudes (80 - 100 km).





Something different is found with regards to MLS - i.e. negative difference at all altitudes, ranging from -40% to -10% - which is in accordance with the stated MLS bias.

Given its unique extension in time and geographical coverage, this new mesospheric CO dataset provides a valuable tool for further studies of mesosphere dynamics.

## 5 Appendix A



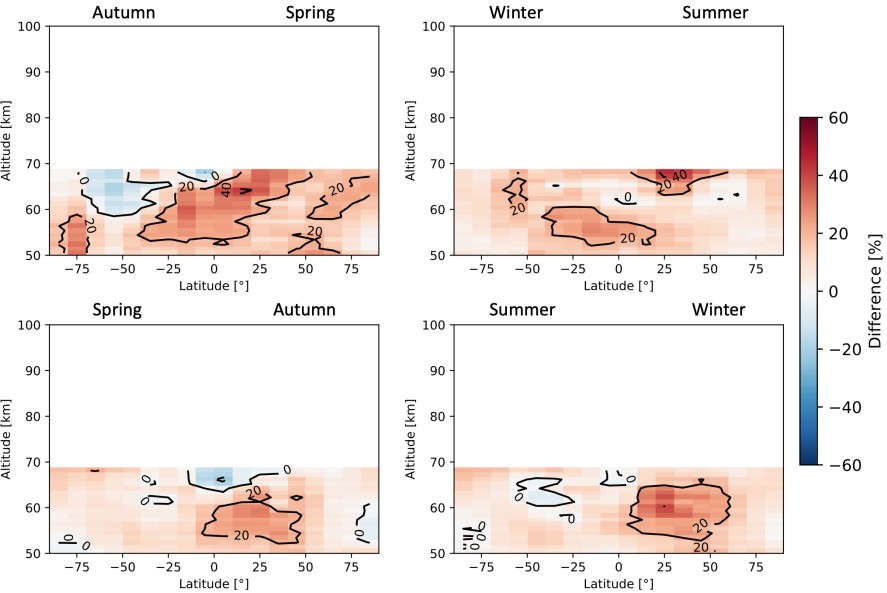

**Figure A1.** Seasonal zonal means of SMR–MIPAS Nominal relative differences averaged over the time period indicated in Table 2. The seasons are intended as the time between solstice and equinox.

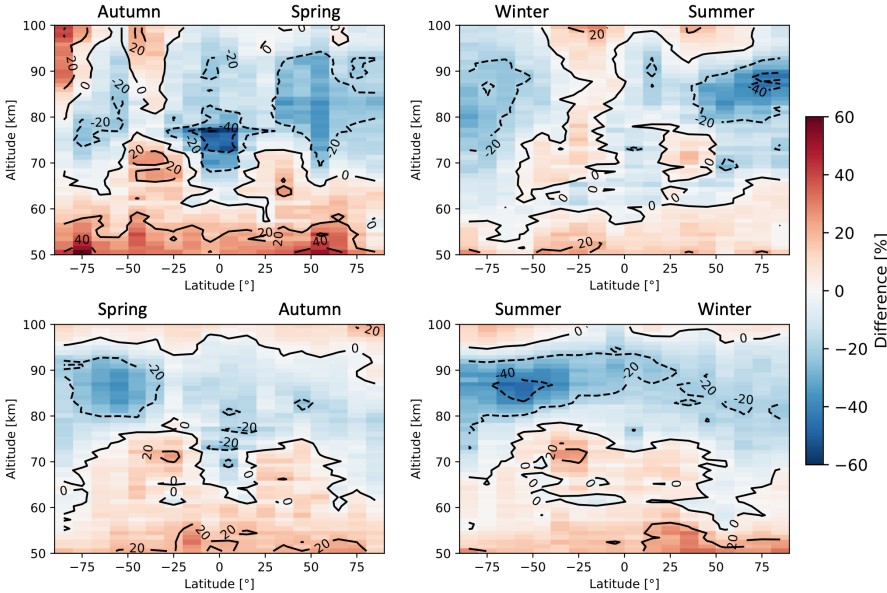

**Figure A2.** Seasonal zonal means of SMR–MIPAS Middle Atmosphere relative differences averaged over the time period indicated in Table 2. The seasons are intended as the time between solstice and equinox.





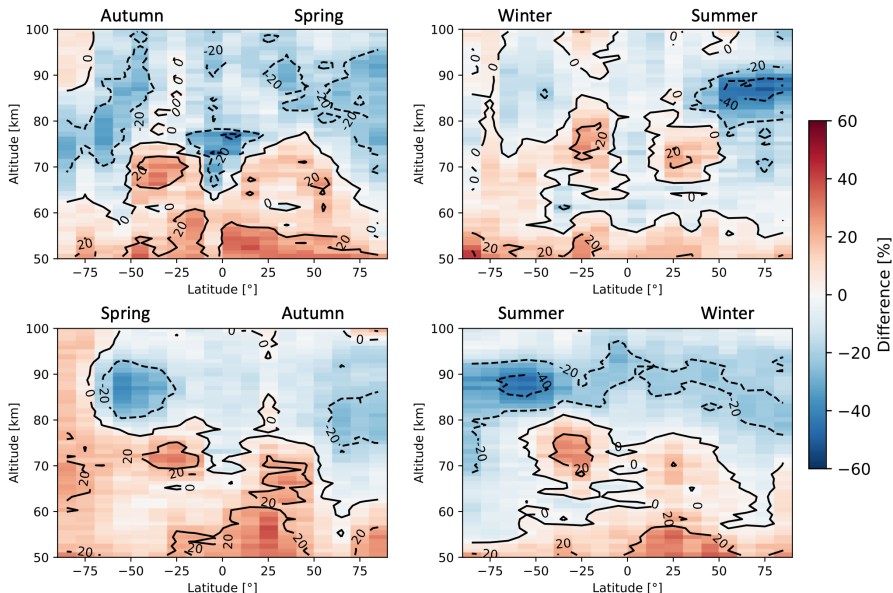

**Figure A3.** Seasonal zonal means of SMR–MIPAS Upper Atmosphere relative differences averaged over the time period indicated in Table 2. The seasons are intended as the time between solstice and equinox.

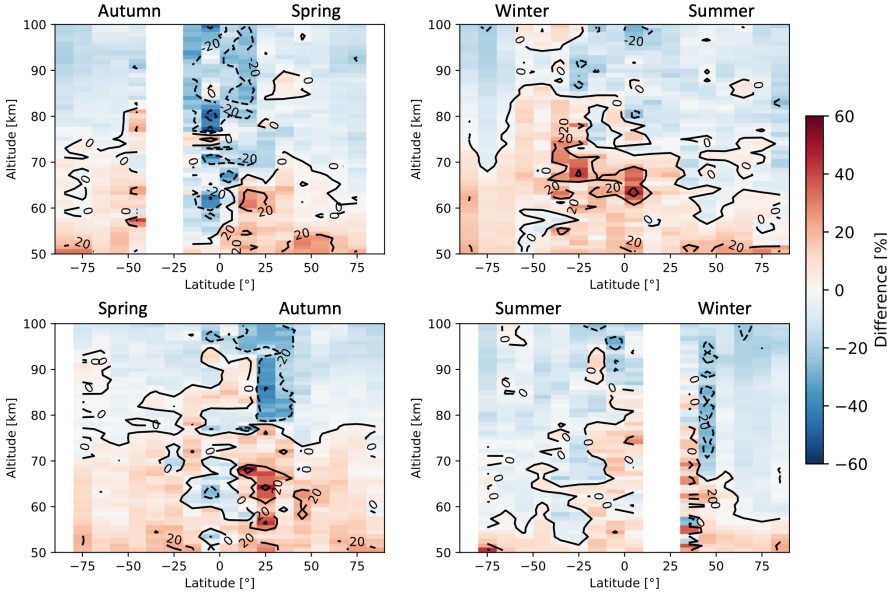

**Figure A4.** Seasonal zonal means of SMR–ACE relative differences averaged over the whole time between February 2004 and March 2019. The seasons are intended as the time between solstice and equinox.

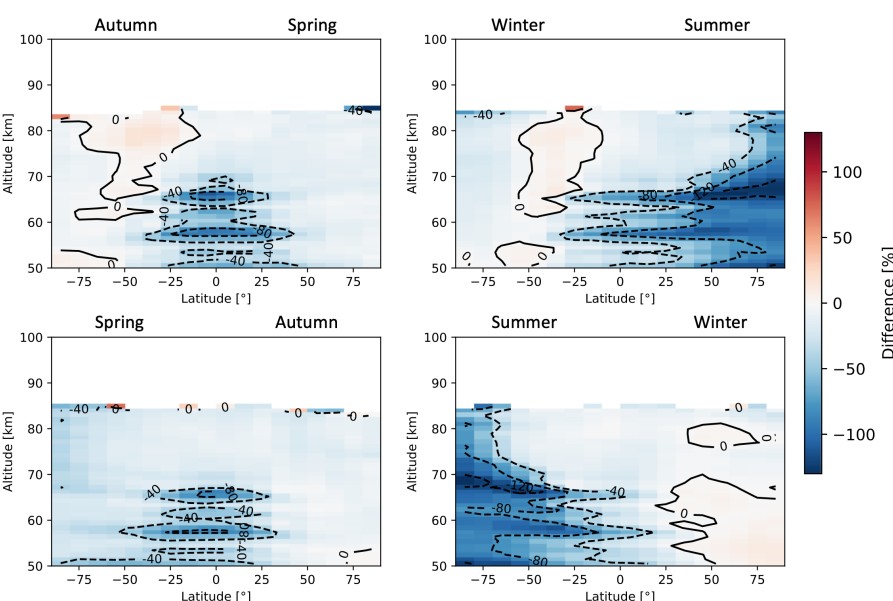

**Figure A5.** Seasonal zonal means of SMR–MLS relative differences averaged over the whole time between July 2004 and March 2019. The seasons are intended as the time between solstice and equinox. Note the different colour scale compared to the previous figures.



*Data availability.* Odin/SMR v3.0 L2 data are publicly accessible at http://odin.rss.chalmers.se/level2; MIPAS IMK/IAA L2 data (both NOM and MA/UA) can be downloaded upon registration at http://www.imk-asf.kit.edu/english/308.php; ACE-FTS L2 data are available upon request at https://database.scisat.ca/l2signup.php; MLS L2 data are available at https://doi.org/10.5067/Aura/MLS/DATA2005 (Schwartz et al., 2015).

*Author contributions.* FG has developed the correction algorithm, processed the CO data, made most of the plots and written most of the text. KP, DM and PE have initiated the project and supported FG throughout it. KP has contributed to the writing of the text. PF has performed the comparison study between SMR and the OSO radiometer and has written the corresponding section. BR has helped with the processing of the data. All co-authors have contributed to the interpretation of the results and have proofread the text.

*Competing interests.* The authors declare that they have no conflict of interest.

*Acknowledgements.* The Chalmers team acknowledges support from the Swedish National Space Agency (Dnr 88/14 and 72/17). Odin is a Swedish-led satellite mission, which is also part of the European Space Agency's (ESA) third party mission programme. The reprocessing of the SMR data was supported by ESA (MesosphEO and Odin/SMR reprocessing projects). The authors would like to thank Julia Ringsby for her contribution to the correction algorithm. BF acknowledges support by the Spanish MCINN (ESP2017-87143-R) and EC FEDER funds. The Atmospheric Chemistry Experiment is a Canadian-led mission mainly supported by the Canadian Space Agency. MLS research
in Edinburgh was funded by NERC.



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
