# Peer review of "Recovery and validation of Odin/SMR long term measurements of mesospheric carbon monoxide"

_Atmospheric Measurement Techniques, 2020_

## Referee Comment (RC1) · Anonymous Referee #1 · 26 Apr 2020

This manuscript outlines the approaches taken to correct previously unusable data from the Sub-Millimeter Radiometer (SMR) on board the Odin satellite, presents results for the altitude range 50-100 km over the period from 2004-2017. It then compares the results, which range over more than 2 orders of magnitude in volume mixing ratio, with data from MIPAS, ACE-FTS and MLS satellite instruments, and the OSO ground station in Sweden. The data show expected seasonal and latitudinal variations. Global mean differences with the other instruments is shown to be in the range +20% to -30%, higher than the comparisons at the lower altitudes and lower at the higher altitudes, especially around 85 km, but with no attempt at an explanation. However, differences in some seasons and latitude regions can show much larger differences, especially

compared to MIPAS, again with no discussion, or comments on how to resolve who is right. The paper would be more complete if it indicated how, with these large seasonal and latitudinal uncertainties, the data can be scientifically used.

Specific Comments:

More information is needed on SMR in Section 2, for instance Are the 2 side bands folded over on each other, or are they separate? (Could be part of more detailed explanation of Fig. 3). Is the scan continuous, or a step scan? Section 3.1: The method looks conceptually straightforward. Is there a reason the this has not been done before? It looks as though the method described will work from 40 to 70 km, but what is done above that? Is the monthly a priori also divided into latitude bins? What is the "measurement response"? Is this the same as the Degrees of Freedom of the Signal (DFS)? Section 3.2: Does the retrieval depend on temperature? If so, what temperature is used? What are its Biases? P. 7, Line 12 (regarding Fig. 5)- isn't it more that the concentrations above 90 km are influenced by the signals there? P. 9, l. 25: Please relate the vertical motions associated with the SAO with these signals l. 28: clarify which is "this latitude band". P. 10, l. 1: Isn't it more accurate to say SSW's occur almost entirely in the N.H. P. 12: l.2: What was the criterion for sufficient coincidences? l. 11: If this is absolute difference, please denote with abs. value brackets. Why use absolute values- aren't you interested in whether the difference is positive or negative? If not, why? ll. 17-18: Wording is unclear- is the average difference being calculated, or the median, or some combination? Is $\bar{\Delta}(z)$ the mean of $\bar{\Delta}_{abs,i}(z)$? Where does the median come in? The SEM expression is the standard for calculating the standard deviation of the mean. P. 13, l. 12: Why were the 2 different periods with different MIPAS spectral resolution lumped together? Does the MIPAS spectral resolution not make a difference? The difference plots in Fig. 9a stop at ∼ 70 km. What is plotted in the right panel above that- is it all a priori? P. 14, Fig. 9: What is the explanation, or theory for, the negative peak in the comparisons ∼ 85 km? Since this occurs for all global comparisons, it would seem that it is a feature of the SMR measurements

and/or analysis. On the same subject, there are very large and fluctuating differences as functions of season and latitude. Do the authors have any explanation for these variations?

Note that the large number of comparisons make the SEM's very small and perhaps not useful.

P. 16, l 3 MLS: Since MLS is another microwave instrument, it is surprising that its data are more different that that from MIPAS or ACE-FTS. The authors should comment on why this might be the case. Is it instrumental, or arising from the data analysis?

P. 17, ll. 4-5: Again, what is measurement response? Why a criterion of 0.75? Were OSO data only available for 2002-2007? The description seems to indicate an instrument improvement, especially after 2014. P. 19, Fig. 14: The difference of the comparisons between MLS and the other instruments is striking, as noted above. Also, the consistent decrease from higher to lower than the comparisons with altitude.

Appendix: Latitudinal and seasonal differences in the biases at high altitudes- who is right?

Looking at Figs. A2 and A3, it strikes this reviewer that the strong differences with MIPAS occur at the summer mesopause, the coldest region of the atmosphere. However, this difference does not show up in the ACE-FTS comparisons. This suggests that the MIPAS CO mixing ratios might be too high, which could result from their temperatures being too low. At this level MIPAS temperatures must take non-LTE effects into account. These are notoriously difficult, and a negative error could lead to CO amounts too high.

Technical Comments:

Page 1,Line 7: Much of the level 1... (The not needed) l. 8: front end l. 17: comparison instruments P2,l. 19,20- Suggest wording- "...obtsined overall agreement over a two order of magnitude agreement" l.22: front end P3, l. 19 downward) scans P4, l. 7 front

end P. 10, Fig. 7- tick marks on abscissa can't be seen, also Fig. 8. P. 15, l. 13: Does SCISAT-1 need to be explained and capitalized? l.24: monotonically, not monotonously P. 16, l. 17: better English would be ...the MLS bias described is...

––––––––––––––––––––––

---

## Referee Comment (RC2) · Anonymous Referee #2 · 12 May 2020

This paper describes the recovery and retrieval of middle atmospheric CO data from the SMR instrument onboard the Odin satellite. It presents a valuable new dataset for the analysis of mesospheric and upper stratospheric transport processes and dynamics. The paper is well written and the figures and tables are general clear and effective at conveying the essential points. There is one major issue on the recovery method that ought to be addressed, along with a number of more minor comments and suggestions. The paper is highly suitable for publication in AMT provided that these issues are addressed.

**Major**

There seems to be two major obstacles to overcome during periods when the PLL was malfunctioning: frequency shifting and line broadening. The approaches taken to overcome these obstacles seem reasonable, but more information must be provided in the paper in order to assess whether this is the case.

1. The paper is not entirely clear about this, but my impression is that from 2001-2003, the PLL was malfunctioning but only a frequency correction was need. In 2003-2004, the PLL was working and no corrections were applied. Then from 2004 onward, the PLL was malfunctioning but both the frequency and broadening corrections were needed. My naive assumption was that the broadening was a result of the frequency shift occurring during the time of a single spectrum, but this seems inconsistent with how the corrections were applied. In any event, the paper needs to have more information to better clarify the malfunctions of the PLL during the pre 2003 and post 2004 time periods.

2. For the frequency correction, a "basic" correction applied to all spectra in a scan is derived using differences in vertical gradients of brightness temperature ($T_b$) between O3 and CO. This is a novel idea and it seems to work well. However, Fig 3 compares $T_b$ profiles for global averages, and one might expect to see differences according to latitude or season since vertical gradients in O3 and CO mixing ratios do change with season and latitude. Whether seasonal or latitudinal variations matter for this "gradient threshold" method is something that needs to be demonstrated. There should also be bars or shading to show the sigma in Fig 3.

3. A second, single-altitude frequency correction seems to be applied using Gaussian fits, but it is not discussed in sufficient detail at the top of p. 6. Why is a Gaussian thought to be the appropriate lineshape (later, thermal broadening is mentioned but what about pressure broadening at the bottom of a scan)? Was the central frequency and half-width fitted independently using some least-squares method? If so, how do the fitted widths compare with those expected? Was this second stage of frequency correction required for both 2001-2003 and post 2004 periods, or only post 2004 as for the broadening correction?

4. Figures 2 and 4 should have color bar legends to show the approximate range of altitudes covered.

5. It is important to show how the frequency shift and broadening varied with time, to give an idea of the magnitudes and temporal variation of both corrections. For example, monthly means of both quantities such as in Fig 7, used for the total number scans, would give the reader this kind of information.

6. Related to number 5 above, in the dataset there should be quality or error flags (or raw information such as mean scan frequency shift and broadening correction) so that users can further filter or weight these data for scientific study. The paper does not discuss any error estimates in the dataset.

**Minor**

1. abstract line 7 "The much of the..." -> "Much of the..."

2. p. 2  l. 3 "Schumann-Runge *bands* and continuum..."

3. p. 2. l. 27 "This is the first data being part of the..." is confusing

4. caption of Fig 3 "...and in the right place." -> "...and at the expected frequency."

5. p. 5 l. 13 "...a value higher than -0.0045 K/m..." Since all slopes are negative, this is confusing. I think what is meant is a slope with a magnitude less than 0.0045 K/m. Clearly, O3 has a larger magnitude in the vertical gradient of T_b, and CO a much smaller vertical gradient in T_b. Referring to slopes in Fig 3 is confusing because CO has a steeper slope, but a smaller vertical gradient.

6. p. 6 l. 7 define ARTS at first use

7. p. 7 l. 10 "Despite *the fact that* spectra above 40 km tangent altitude are considered..."

8. p. 8 l. 6 Is the given sigma FWHM or HWHM?

9. p. 10 l. 1 High CO vmr due to enhanced descent at the stratopause following an SSW was clearly demonstrated by Manney et al (Atmos. Chem. Phys., 9, 4775–4795, 2009).

10. p. 12, l. 5-10 Not enough attention is given to possible biases due to differences in vertical resolution. There can be large vertical gradients in CO that depend on altitude and season. Between 70km and 90 km, where some biases are shown, CO increases by an order of magnitude over 20 km and differences in vertical averaging could be a factor. The mean averaging kernel widths between the various instruments should be given. For example, it is possible that ACE has at least a factor or 2 higher vertical resolution than SMR.

---

## Author Comment (AC1) · 7 Jul 2020

More information is needed on SMR in Section 2, for instance Are the 2 side bands folded over on each other, or are they separate? (Could be part of more detailed explanation of Fig. 3). Section 2.1 has been updated with the following: "The output signal of the mixer is a function of the sum and difference of the frequencies of the input signals, therefore consisting of two sidebands. SMR is however run in single sideband mode so, of the two sidebands, only the one contaning the signal of interest should be transmitted and the other one suppressed. This is achieved, before mixing, by accordingly setting the length of the arms of the Martin-Puplett interferometer."

Is the scan continuous, or a step scan? It is a continuous scan with a constant speed of 0.75 km/s. This information has been added to Section 2.1.

Section 3.1: The method looks conceptually straightforward. Is there a reason the this has not been done before? The reason was a lack of personnel that would take care of this.

It looks as though the method described will work from 40 to 70 km, but what is done above that? Once the line has been identified via the spectra in the 40 - 60 km range, all the spectra in the scan are moved in bulk to the theoretical position. This is done by calculating the mean spectrum of all spectra in the scan, then the center frequency of the line is identified and the difference to the theoretical center is calculated. This same value is applied as a frequency correction for all spectra in the scan. After that a second correction, different for each spectrum, is applied, as described in Section 3.1 of the paper. We have clarified that point in Section 3.1.

Is the monthly a priori also divided into latitude bins? Yes, it is divided into 10° latitude bins. This information has been added to the text.

What is the "measurement response"? Is this the same as the Degrees of Freedom of the Signal (DFS)? The following was added to the text: ", where the measurement response is a measure of the degree to which the result may be contaminated by the a priori. It is defined as the sum over the row of the averaging kernel matrix (Rodgers, 2000)". Also, the measurement response profile was added to Fig.5.

Section 3.2: Does the retrieval depend on temperature? If so, what temperature is used? What are its Biases? Yes, the retrievals depend on temperature. ERA-interim reanalysis data (Dee et al., 2001) is used up to 60 km and the Mass Spectrometer Incoherent Scatter model (version NRLMSISE-00; Picone et al., 2002) is used from 70 km upwards. Between 60 - 70 km, a spline interpolation of the two is applied. Information added to the text.

P. 7, Line 12 (regarding Fig. 5)- isn't it more that the concentrations above 90 km are influenced by the signals there? Yes, concentrations between 90 - 100 km are influenced by the signals there. In the text it is written that they are *also* influenced by higher altitudes measurements, as can be seen by the shape of the averaging kernels.

P. 9, 1. 25: Please relate the vertical motions associated with the SAO with these signals. We have added the following explanation to the manuscript: "Although the SAO-induced variations in zonal mean vertical wind are too small to be measured directly, the accumulated effects can be seen in the distribution of the long-lived constituents like CO (Hamilton,

2015).Sinking of air, or at least anomalously weak rising motion, leads to the observed increase in the CO mixing ratiosaround the equinoxes."

1. 28: clarify which is "this latitude band". This has been changed to "high latitudes".

P. 10, l. 1: Isn't it more accurate to say SSW's occur almost entirely in the N.H. We have changed "occur mainly in the N.H." to "occur almost entirely in the N.H."

P. 12: 1.2: What was the criterion for sufficient coincidences? Coincidences are considered sufficient if they reach at least the number of 10. This information was added to the text.

l. 11: If this is absolute difference, please denote with abs. value brackets. Why use absolute values- aren't you interested in whether the difference is positive or negative? If not, why? Absolute difference just means the subtraction  $x_SMR - x_comp$  (without any normalization, as it is done for the relative difference), which of course can be negative, as it can be seen in the figures showing the comparisons. Absolute difference doesn't mean "the absolute value of the difference" as the reviewer suggests.

ll. 17-18: Wording is unclear- is the average difference being calculated, or the median, or some combination? Is ïA D(z) the mean of ïA d'abs,i (z)? Where does the median come in? We changed the text to "the median difference Delta(z) ... is calculated."

The SEM expression is the standard for calculating the standard deviation of the mean. True, it is the same as for the standard deviation of the mean, only that here Delta(z) is the median, not the mean.

P. 13, l. 12: Why were the 2 different periods with different MIPAS spectral resolution lumped together? Does the MIPAS spectral resolution not make a difference? The reviewer is right. The comparison has been re-done to consider the two periods separately and the paper has been updated accordingly. Figures 9 and 14 have been updated to show comparison with FR-NOM and OR-NOM separately. The following has been added to the text: "Regarding FR-NOM mode, with the above mentioned coincidence criteria we found 6088 coincident measurements with SMR over the period July 2002 - March 2004. The profiles and differences between 50 and 70 km altitude, averaged over the whole time period and over the whole globe, are shown in Figure 9a. The median relative difference between SMR and MIPAS has a value of +8% at 50 km which decreases to reach -15% at 60 km, and eventually remains constant between 60 - 70 km altitude. No bias characterising a particular latitude or season over time has been identified for this specific comparison, since FR-NOM mode has been operational for only a short period.

Comparing with OR-NOM mode, 88902 coincident measurements have been found over the period January 2005 - April 2012. Figure 9b shows that the median relative difference between SMR and MIPAS remains positive and almost constant with altitude, staying below +15%. The main contribution to this difference comes from latitudes between -25° and +50°. In particular during northern summer, around +25°, where it reaches values above +40% between 65 and 70 km. Everywhere else relative difference values remain within  $\pm$  20% (see Figure A1)."

The difference plots in Fig. 9a stop at  $\sim$  70 km. What is plotted in the right panel above thatis it all a priori? Nothing is plotted above 70 km in the right panel, as well as in the center panel. In fact, differences between the two instruments are plotted in these panels and, since MIPAS nominal data extend only up to 70 km, there can be no difference shown above that altitude. In case the reviewer meant the left panel, note that what is plotted above 70 km is SMR CO concentration which is retrieved up to 100 km. So no, it is not all a priori.

P. 14, Fig. 9: What is the explanation, or theory for, the negative peak in the comparisons  $\sim$ 85 km? Since this occurs for all global comparisons, it would seem that it is a feature of the SMR measurements and/or analysis. On the same subject, there are very large and fluctuating differences as functions of season and latitude. Do the authors have any explanation for these variations? As explained more extensively in the dedicated points, SMR-MLS differences are consistent with a known bias of MLS (positive systematic error of 20-50%). Regarding concentration differences with other instruments we have checked that they are not due to differences in vertical resolution. This has been done by smoothing SMR profiles with a Gaussian filter to account for differences in vertical resolutions; however, this smoothing didn't lead to significative changes in the profile. Also, we isolated MIPAS daytime profiles (which have a vertical resolution comparable to SMR) but the same relative differences with SMR were observed as in the globally averaged profiles. Moreover, it is unlikely that SMR-MIPAS relative differences are due to a misestimation of non-LTE effects from MIPAS. since systematic errors due to non-LTE were estimated to be less then 5% (in Funke et al., 2007, doi:10.1029/2006JD007933). This being said, the causes for the observed concentration differences, both globally and for different latitudes and seasons, are unknown. This information has been added to the Section "Conclusion" of the paper.

Note that the large number of comparisons make the SEM's very small and perhaps not useful. It is true that the SEM is sometimes not visible, but the fact that it is very small is a useful information. Moreover, there are some cases where it is not negligeable and, for the sake of consistency, we should indicate it in all comparisons.

P. 16, 13 MLS: Since MLS is another microwave instrument, it is surprising that its data are more different that that from MIPAS or ACE-FTS. The authors should comment on why this might be the case. Is it instrumental, or arising from the data analysis? It is a known bias of MLS, already referred to in the text in Section 5.3. MLS CO, as reported e.g. in Errera et al.(2019) and Livesey et al.(2018), is known to be affected by a positive systematic error of 20–50% throughout the mesosphere. This explains the observed SMR-MLS differences. Moreover, although SMR and MLS are both microwave instruments, they are not using the same spectral line, so there may be a variety of spectroscopic errors which would affect one instrument and not the other. However, CO is a species that radio astronomers have been studying for many decades; we rather suspect that its spectroscopy is better known than that of most other molecules. When MLS validation has been performed in Pumphrey et al. (2007), CO spectroscopy revealed itself to cause a 10% effect at worst; less serious than a variety of other systematic errors.

P. 17, ll. 4-5: Again, what is measurement response? Why a criterion of 0.75? For definition see point above. The value of 0.75 has been used for a long time. The reason is that, using a higher value as a threshold would result in the filtering out of a great part of the data.

Were OSO data only available for 2002-2007? The description seems to indicate an instrument improvement, especially after 2014. Most of the OSO data after 2014 has not been processed yet and the corresponding L2 data is therefore not available. The instrument improvement has been described just to shortly point out that OSO does not operate with a single sideband mixer anymore.

P. 19, Fig. 14: The difference of the comparisons between MLS and the other instruments is striking, as noted above. Also, the consistent decrease from higher to lower than the comparisons with altitude. As mentioned above, MLS CO is known to be characterized by a positive systematic error of 20–50% throughout the mesosphere.

Appendix: Latitudinal and seasonal differences in the biases at high altitudes- who is right? The causes for the observed concentration differences are unknown.

Looking at Figs. A2 and A3, it strikes this reviewer that the strong differences with MIPAS occur at the summer mesopause, the coldest region of the atmosphere. However, this difference does not show up in the ACE-FTS comparisons. This suggests that the MIPAS CO mixing effects might be too high, which could result from their temperatures being too low. At this level MIPAS temperatures must take non-LTE effects into ac- count. These are notoriously difficult, and a negative error could lead to CO amounts too high.

1) non-LTE modelling of MIPAS CO has been extensively validated in Funke et al., 2007 (doi:10.1029/2006JD007933) and systematic errors due to non-LTE were estimated to be less then 5%. This information has been added to Section 5.1.

2) MIPAS CO (at least during daytime and hence polar summer) is rather independent on kinetic temperature, since CO daytime vibrational temperatures are dictated by solar excitation. A mapping of T errors is thus very unlikely, except for indirect effects via hydrostatics.

3) Further, the MIPAS kinetic temperature product used in the CO retrievals has been extensively validated in Garcia-Comas et al, 2014 (https://www.atmos-meas-tech.net/7/3633/2014/) and differences to correlative measurements were found to be less than 4K in the polar summer mesopause region.

4) the polar summer mesosphere, where the most striking differences occur, is characterised by very large vertical gradients in the CO distribution. In consequence, differences in vertical resolution between both instruments are likely important, there. While the vertical resolution of mesospheric MIPAS daytime CO is around 5 km (similar to that of SMR), nighttime vertical resolution is worse (>10 km) because of the lower sensitivity caused by smaller nighttime non-LTE emissions (no solar excitation). Therefore, it could be possible, that the SMR-MIPAS comparison is affected by the lower MIPAS nighttime resolution. This has been checked by restricting the comparison to MIPAS daytime profiles, but the resulting difference profiles are very similar to the ones considering data altogether. This suggests that differences in vertical resolution are not the cause of the observed biases. This information has been added to Section 5.1.

Technical Comments:

Page 1,Line 7: Much of the level 1... (The not needed) OK

1. 8: front end OK

1. 17: comparison instruments OK

P2,l. 19,20- Suggest wording- ". . .obtsined overall agreement over a two order of magnitude agreement" Not applied. We think the current wording is appropriate.

1.22: front end OK

P3, l. 19 downward) scans Not applied. We would like to avoid using the word "scan" in the definition of scan.

P4, l. 7 front end OK

P. 10, Fig. 7- tick marks on abscissa can't be seen, also Fig. 8. OK

P. 15, l. 13: Does SCISAT-1 need to be explained and capitalized? The name of the satellite is written in lowercase on the ESA website so we don't think it needs to be written in capital letters.

1.24: monotonically, not monotonously OK

P. 16, l. 17: better English would be . . . the MLS bias described is. . . OK

---

## Author Comment (AC2) · 7 Jul 2020

**Major**

There seems to be two major obstacles to overcome during periods when the PLL was malfunctioning: frequency shifting and line broadening. The approaches taken to overcome these obstacles seem reasonable, but more information must be provided in the paper in order to assess whether this is the case.

1. The paper is not entirely clear about this, but my impression is that from 2001-2003, the PLL was malfunctioning but only a frequency correction was need. In 2003-2004, the PLL was working and no corrections were applied. Then from 2004 onward, the PLL was malfunctioning but both the frequency and broadening corrections were needed. My naive assumption was that the broadening was a result of the frequency shift occurring during the time of a single spectrum, but this seems inconsistent with how the corrections were applied. In any event, the paper needs to have more information to better clarify the malfunctions of the PLL during the pre 2003 and post 2004 time periods. Section 3.3 has been updated to better clarify that we think two different kinds of PLL malfunctioning affected the instrument during these two periods: one before 8 October 2003 which had effect only at longer time scales and caused only frequency shifts; and another one after 8 October 2004 which have effects both within the integration time and at longer time scales, causing both line broadenings and frequency shifts.

2. For the frequency correction, a "basic" correction applied to all spectra in a scan is derived using differences in vertical gradients of brightness temperature (T\_b) between O3 and CO. This is a novel idea and it seems to work well. However, Fig 3 compares T\_b profiles for global averages, and one might expect to see differences according to latitude or season since vertical gradients in O3 and CO mixing ratios do change with season and latitude. Whether seasonal or latitudinal variations matter for this "gradient threshold" method is something that needs to be demonstrated. There should also be bars or shading to show the sigma in Fig 3. A panel which represents the fitted values of the slopes together with uncertainties has been added to Figure 3 and is described in Section 3.1. In this panel it can be seen how the difference between the two slopes is bigger than any natural variability. We therefore confirm that the threshold we are using is valid for whatever season and latitude.

3. A second, single-altitude frequency correction seems to be applied using Gaussian fits, but it is not discussed in sufficient detail at the top of p. 6. Why is a Gaussian thought to be the appropriate lineshape (later, thermal broadening is mentioned but what about pressure broadening at the bottom of a scan)? Was the central frequency and half-width fitted independently using some least-squares method? If so, how do the fitted widths compare with those expected? Was this second stage of frequency correction required for both 2001-2003 and post 2004 periods, or only post 2004 as for the broadening correction? The Gaussian fits which are discussed here are not to be confused with the ones performed during inversions to retrieve concentration profiles. Here we are using Gaussian fits exclusively as a method to estimate the center frequency of the line so to assess what is the frequency correction that needs to be applied to the spectrum. The fitted width, in this case, is of no interest. This is done, prior to the inversion process, for all scans. Another method to obtain the same information could have been to locate the position of the maximum peak of the line. But the use of one method or the other didn't present significant differences in the estimated centrum. In fact, even where the line is not Gaussian, the central part can still reasonably be fitted with one and the obtained center will be correct.

4. Figures 2 and 4 should have color bar legends to show the approximate range of altitudes covered. The paper has been updated with legends reporting the colours corresponding to some of the altitudes. Not all altitudes are indicated in the legend for the sake of readability.

5. It is important to show how the frequency shift and broadening varied with time, to give an idea of the magnitudes and temporal variation of both corrections. For example, monthly means of both quantities such as in Fig 7, used for the total number scans, would give the reader this kind of information. As described in Section 3.3, the broadening correction applied was one and the same for the whole period after 8 October 2004. The sigma value chosen for the new response function is the best compromise over this period, since it is the one giving the lowest overall area differences. Thus, the broadening due to the PLL malfunction was assumed to have the same entity over the whole period after 8 October 2004. The period before instead was not affected by any broadening. Regarding frequency shift, it appeared to be random and not follow any trend with time. We also observed no dependency on the satellite temperature. Therefore, we chose not to include any time series reporting the temporal variations of the corrections. This information has been added in Section 3.1.

6. Related to number 5 above, in the dataset there should be quality or error flags (or raw information such as mean scan frequency shift and broadening correction) so that users can further filter or weight these data for scientific study. The paper does not discuss any error estimates in the dataset. ARTS retrieval algorithms are based on the Optimal Estimation Method (OEM), thus statistical errors come from there (information added to the paper in Section 3.2). Moreover, in Section 3.1 we already explain how we discard spectra were CO line is not present. For all the other spectra, the PLL-generated frequency shifts and line broadening have been corrected, so there is no need of data flags.

**Minor**

1. abstract line 7 "The much of the..." -> "Much of the..." OK

2. p. 21. 3 "Schumann-Runge \*bands\* and continuum..." OK

3. p. 2. l. 27 "This is the first data being part of the..." is confusing Changed to "they are part of the..."

4. caption of Fig 3 "...and in the right place." -> "...and at the expected frequency." OK

5. p. 5 l. 13 "...a value higher than -0.0045 K/m..." Since all slopes are negative, this is confusing. I think what is meant is a slope with a magnitude less than 0.0045 K/m. Clearly, O3 has a larger magnitude in the vertical gradient of T\_b, and CO a much smaller vertical gradient in T\_b. Referring to slopes in Fig 3 is confusing because CO has a steeper slope, but a smaller vertical gradient. A panel which represents the fitted values of the slopes together with uncertainties has now been added to Figure 3. With the addition of this panel the text is not confusing anymore and it has been left unchanged.

6. p. 6 l. 7 define ARTS at first use OK

7. p. 7 l. 10 "Despite \*the fact that\* spectra above 40 km tangent altitude are considered..." OK

8. p. 8 l. 6 Is the given sigma FWHM or HWHM? Neither of the two. Sigma is the standard deviation of the Gaussian curve. The relation between sigma and FWHM is given by: FWHM

= 2\*sqrt(2\*ln(2))\*sigma. We have not modified the text since this simply corresponds to the standard definition of a Gaussian function.

9. p. 10 l. 1 High CO vmr due to enhanced descent at the stratopause following an SSW was clearly demonstrated by Manney et al (Atmos. Chem. Phys., 9, 4775–4795, 2009). Reference added.

10. p. 12, 1. 5-10 Not enough attention is given to possible biases due to differences in vertical resolution. There can be large vertical gradients in CO that depend on altitude and season. Between 70km and 90 km, where some biases are shown, CO increases by an order of magnitude over 20 km and differences in vertical averaging could be a factor. The mean averaging kernel widths between the various instruments should be given. For example, it is possible that ACE has at least a factor or 2 higher vertical resolution than SMR. Regarding comparisons with MIPAS, the polar summer mesosphere, where there most striking differences occur, is indeed characterised by very large vertical gradients in the CO distribution. In consequence, differences in vertical resolution between both instruments are likely important, there. While the vertical resolution of mesospheric MIPAS daytime CO is around 5 km (similar to that of SMR), nighttime vertical resolution is worse (>10 km) because of the lower sensitivity caused by smaller nighttime non-LTE emissions (no solar excitation). Therefore, it could be possible, that the SMR-MIPAS comparison is affected by the lower MIPAS nighttime resolution. This has been checked by restricting the comparison to MIPAS daytime profiles, but the resulting difference profiles are very similar to the ones obtained considering data altogether. This information has been added to Section 5.1.

Moreover, we smoothed SMR and comparison instruments profiles using Gaussian filters with various FWHMs. This has been done in order to account for differences in vertical resolution, but the test didn't show any improvement in concentration differences. This suggests that differences in vertical resolution are not the cause of the observed biases. This information has been added to the Section "Conclusion" of the paper.

---

## Author Response (AR2)

Page 7, line 28: "scanis" -> "scan is" OK
Page 9, line 18: "from there on" -> "from then on" OK
Page 10, line 3: "cover the whole globe until the polar regions". "Until" is unclear and may mean "including the polar regions" or "excluding the polar regions". Perhaps a simple statement of the latitude range involved would also be useful. Text modified from "
[revised manuscript text omitted]